# Feedback of individual genetic and genomics research results: A qualitative study involving grassroots communities in Uganda

Joseph Ochieng[1,2]*, Betty Kwagala[3], John Barugahare[4], Marlo Möller[5], Keymanthri Moodley[2]

**1** Makerere University School of Biomedical Sciences, Kampala, Uganda, **2** Centre for Medical Ethics and Law, Faculty of Medicine and Health Sciences, Stellenbosch University, Stellenbosch, South Africa, **3** Makerere University School of Statistics and Planning, Kampala, Uganda, **4** Makerere University School of Humanities, Kampala, Uganda, **5** Division of Molecular Biology and Human Genetics, DSI-NRF Centre of Excellence for Biomedical Tuberculosis Research, South African Medical Research Council Centre for Tuberculosis Research, Faculty of Medicine and Health Sciences, Stellenbosch University, Cape Town, South Africa

* ochiengjoe@yahoo.com

**Data Availability Statement:** All relevant data are within the paper and its Supporting information files.

## Abstract

### Background

Genetics and genomics research (GGR) is associated with several challenges including, but not limited to, methods and implications of sharing research findings with participants and their family members, issues of confidentiality, and ownership of data obtained from samples. Additionally, GGR holds significant potential risk for social and psychological harms. Considerable research has been conducted globally, and has advanced the debate on return of genetic and genomics testing results. However, such investigations are limited in the African setting, including Uganda where research ethics guidance on return of results is deficient or suboptimal at best. The objective of this study was to assess perceptions of grassroots communities on if and how feedback of individual genetics and genomics testing results should occur in Uganda with a view to improving ethics guidance.

### Methods

This was a cross-sectional study that employed a qualitative exploratory approach. Five deliberative focus group discussions (FGDs) were conducted with 42 participants from grassroots communities representing three major ethnic groupings. These were rural settings and the majority of participants were subsistence farmers with limited or no exposure to GGR. Data were analysed through thematic analysis, with both deductive and inductive approaches applied to interrogate predetermined themes and to identify any emerging themes. NVivo software (QSR international 2020) was used to support data analysis and illustrative quotes were extracted.

### Results

All the respondents were willing to participate in GGR and receive feedback of results conditional upon a health benefit. The main motivation was diagnostic and therapeutic benefits as

**Funding:** Research reported in this publication was supported by the National Human Genome Research Institute of the National Institutes of Health under Award Number U01HG009822 and NIH Fogarty grant under Award Number D43TW01511. The content is solely the responsibility of the authors and does not necessarily represent the official views of the National Institutes of Health. The funder had no role in the data collection and analysis, decision to publish or preparation of the manuscript.

**Competing interests:** The authors have declared that no competing interests exist.

**Abbreviations:** CE, Community Engagement; FGDs, Focus Group Discussions; GGR, Genetics and Genomics Research; REC, Research Ethics committee.

well as facilitating future health planning. Thematic analysis identified four themes and several sub-themes including 1) the need-to-know health status 2) paternity information as a benefit and risk; 3) ethical considerations for feedback of findings and 4) extending feedback of genetics findings to family and community.

## Conclusion

Participation in hypothetical GGR as well as feedback of results is acceptable to individuals in grassroots communities. However, the strong therapeutic and/or diagnostic misconception linked to GGR is concerning given that hopes for therapeutic and/or diagnostic benefit are unfounded. Viewing GGR as an opportunity to confirm or dispute paternity was another interesting perception. These findings carry profound implications for consent processes, genetic counselling and research ethics guidance. Privacy and confidentiality, benefits, risks as well as implications for sharing need to be considered for such feedback of results to be conducted appropriately.

## Introduction

Although the expanding applicability of knowledge generated from (GGR) holds great promise for discoveries in the biomedical and socio-behavioural sciences, it also raises challenging ethical and societal issues. Such challenges include, but are not limited to, implications of sharing research findings with participants and their family members, issues of confidentiality and determining appropriate strategies for providing information to individuals tested [1–3]. Furthermore, GGR has significant potential risk for social and psychological harms. For example, studies that generate information about an individual's health risks can provoke anxiety and confusion, damage familial relationships due to misaligned paternity, and/or compromise the individual's future financial status [4–7]. Results could also possibly be used as a basis for ethnic/racial segregation or discrimination such as denial of insurance coverage or employment [8].

A significant amount of research has been conducted in the fields of genetics and genomics. Associated findings have contributed to the global debate on return of GGR results [9–14]. Although international policies for return of individual genetic research findings are still evolving, there is general consensus for feedback of results. A number of criteria need to be met including 1) the ability to assess the evidence base for potentially disease-causing genetic variants in relation to the concerned population(s); 2) assessment of whether the particular finding is beneficial to the individual; and 3) ensuring that patients are appropriately informed of the implications of the findings for their disease or treatment, and referral for follow-up care while seeking guidance of the Research Ethics Committee (REC) [15]. However, debate addressing similar issues of relevance to the African setting [1, 2, 16–20], and Uganda in particular is still limited. This situation is exacerbated by the fact that many countries in the African region including Uganda lack ethics guidance on return of results [21].

GGR has been conducted for about 20 years in the Ugandan setting and is expected to continue to increase owing to its potential for advancing targeted disease detection and interventions for both communicable and non-communicable diseases in this resource-limited setting [22]. However, there is a paucity of knowledge on the ethical, legal and social challenges that accompany GGR in the country [23–26]. There are few publications on perspectives of

researchers [24, 26] and research participants [25]. This literature highlighted the need for adequate informed consent, community engagement, genetic counselling, feedback of beneficial/actionable GGR findings and ethics guidance for research. In addition to this list, research participants expressed the need for more effective support during and after participation and as well as feedback of all findings since consideration of beneficial/actionable findings to a good extent is individual based. Published literature on perspectives of grassroots communities is lacking. These communities are based in rural, communal settings with limited interaction with the outside world, are less formally educated and represent lower socio-economic groups. Adding the views of grassroots communities to those of other stakeholders will provide a more comprehensive context for guideline development and conduct of GGR in Uganda.

We set out to assess the perceptions of grassroots communities on feedback of individual GGR results in Uganda to inform development of contextualized research ethics guidelines.

## Methods

### Study design and setting

This was a cross-sectional study that employed a qualitative exploratory approach. The study was conducted by a multi-disciplinary team of researchers comprising social scientists, bioethicists and medical scientists with experience in qualitative research. JO a male medical doctor and academic with bioethics training and experience, BK a female PhD sociology academic of more than 20 years and JB a male PhD Philosophy academic led most of the focus group discussions (FGDs). They were assisted by eight research assistants (four women and four men) who were proficient in the respective local languages. Data were collected between January and February 2021. Participants were recruited from remote grassroots communities in three regions of Uganda each representing a major ethnic grouping. Two deliberative FGDs were conducted in each region with one involving youth (those18-35 years as classified in Uganda) and the other involving individuals of 36 years and above. However, in one of the regions only one FGD involving individuals older than 35 years was conducted. The communities were selected from the eastern, northern and west Nile regions of Uganda to represent the main ethnic groupings. Participants were recruited from predetermined ethnic groups, districts and sub-counties. The specific local communities were selected by the research assistants identified at the respective sub-counties. The research assistants worked with a local mobilizer to identify the potential participants and invited them for the meeting with the study team. Individuals who responded to the call and met the inclusion criteria were given information about the study and those who consented were recruited.

### Data collection

The FGDs were conducted in open spaces in the compounds of health facilities, schools or churches a safe distance away from non-participants to ensure privacy. Data were collected in adherence to the Covid-19 prevention measures which included hand sanitization, face masking, social distancing and utilization of open spaces.

Data collection entailed face to face deliberative FGDs that were conducted in the respective local languages of the selected communities. The discussions took about one and a half hours excluding the education session. Participants were first asked general questions on awareness and knowledge about genetics and genomics. This was followed by a 30-minute explanatory session on the meaning and role of genetic inheritance, DNA, genes, genome and how they affect individual's inherited traits and susceptibility or resistance to some health conditions. Additionally, explanation of genomics research and genomics as well as the testing and feedback of GGR results was done. This education session was followed by a discussion moderated

by the FGD guide. The discussion included willingness to participate in GGR, willingness to receive feedback of GGR results, conditions for feedback and extending feedback to family and community. The discussions were audio recorded and complemented by notes taken by research assistants.

## Data management and analysis

Recorded information was transcribed verbatim, checked for accuracy and later translated into English. Data were analysed along the main themes of the study. The analysis was conducted using a comprehensive thematic matrix that included identifying codes to determine common patterns arising from the narratives. Thematic analysis was done both deductively based on the study predetermined themes and, inductively to identify emerging themes. We started deductively with a set of themes, but then worked inductively to come up with any new themes as we iteratively read through the data [27, 28]. Transcripts were further reviewed for emerging themes which were integrated into the thematic matrix. The researcher, JO was involved in applying and confirming application of codes across all transcripts and disagreements were resolved by cross checking with the recorded data. NVivo software (QSR international 2020) was used to support data analysis and illustrative quotes were extracted.

## Ethical considerations

Ethical review and approval was obtained from the Makerere University School of Biomedical Sciences Higher Degrees and Research Ethics Committee ref. SBS 628 and the Health Research Ethics Committee, Stellenbosch University ref. HREC 16853, followed by clearance by the Uganda National Council for Science and Technology. Ref. SS268ES.

Both men and women aged 18 years and above who had provided written informed consent participated in the study. For non-literate participants, the consent process was witnessed by a literate individual of the participants' choice who was independent from the study and the written consent was documented by a thumb print of the participant followed by signature of the witness. No participant identifying information was recorded.

## Findings

Five deliberative FGDs involving 42 participants were conducted across the three regions of the country. Over half of the FGD participants were male and aged between 18–77 years. The majority were small scale farmers, Christians, married and had children. They all lived in rural communities.

All the participants were willing to participate in GGR and receive feedback of genetic results. The main reasons for receiving genetic results were the need to know one's health status and to seek care or to plan for one's future and the future of their families. However, the willingness to participate in GGR was conditional based on the expectation that feedback of results would occur. Other expectations included adequate informed consent, genetic counselling, implications of testing, privacy and confidentiality.

Thematic analysis identified four themes and a number of sub-themes including 1) the need to know one's health status; with subthemes of therapeutic and/ or diagnostic misconception, as well as concerns, challenges and implications for sharing; 2) paternity information as a benefit and risk; 3) ethical considerations for feedback of findings, with sub-themes of adequate informed consent, genetic counselling as well as privacy and confidentiality; and 4) extending feedback of genetics findings to family and community, (Table 1).

Table 1. Summary of the identified themes.

| Number | Theme | Sub-theme |
|---|---|---|
| 1. | The need to know one's health status | a. Therapeutic and/or diagnostic misconception<br><br>b. concerns, challenges and implications for sharing |
| 2. | Paternity information as a benefit and risk | |
| 3. | Ethical considerations for feedback of genetics findings | a. Adequate informed consent<br><br>b. Genetic counselling<br><br>c. Privacy and confidentiality |
| 4. | Extending feedback of genetics findings to family and community | |

Although the respondents were agreeable to feedback of all GGR results including aggregate results, individual results, incidental findings and secondary findings, for this paper, we focused on perspectives regarding feedback of individual genetics results.

## 1. The need to know one's health status

Almost all the participants responded in the affirmative when asked about their willingness to receive feedback of hypothetical genetic test results because they felt that it was useless to take the test if they would not receive results. All respondents stated that genetic testing is acceptable and would contribute to knowledge in the field. Respondents also felt that findings of such testing need to be shared with individuals tested so that they would be aware of their health status. Knowing one's genetic information was a major motivating factor for participating in GGR or taking a genetic test.

"*When I go for the test, I do so because I want to know my health status, so if I am tested and they don't give me my results it is almost like I have not done any tests, so if I get tested my results should be brought back so that in case I have any underlying conditions I can seek help.*"

***FGD 007 Respondent 4***

"*I want to know my results . . . so that I can know if I am healthy or sick.*"

***FGD 008 Respondent 12***

"*Yes, the results should be provided to the patient. . .. so that I can know exactly what they have found out.*"

***FGD 007 Respondent 1***

Respondents felt that for any test carried out, the results will either turn out to be positive or negative, and for any underlying condition the results will turn out positive meaning that feedback helps one to start living a new life. Respondents noted that even if the treatment may not be available for the diagnosed condition, it would still help them understand their health condition and plan for their future. Thus, if results are not shared, the individuals tested will remain unsettled and anxious wondering what could be happening to their bodies. Some thought that if treatment is not available at the testing centre, it could as well be sought from other hospitals provided one knew what their health problem was.

"*I want to know the results of those tests because I want to know my health status so that if am sick, I go to the hospital, if I am not, I start planning my life afresh.*"

**FGD 008 Respondent 9**

"*It is right because it helps me to know my status which gives me the strength to take care of my children.*"

**FGD 008 Respondent 4**

A minority of respondents wished to know their test results only if the condition is treatable. Otherwise, it would be stressful and cause unnecessary anxiety for one to be told of a disease that has no treatment option.

"*Using my body parts, I am not interested. If they tell me my blood group, I understand but if it is something else, I am not interested.*"

**FGD 007 Respondent 10**

Respondents had various reasons for wanting to know the results of their hypothetical genetics testing including being able to plan for the future, knowing their health conditions and being able to resolve some of the community myths particularly following death of individuals. For others, since the samples were from their bodies, they felt they had a right to know the outcomes of the tests through provision of feedback. Some felt that knowing the results would be helpful in guiding the individuals on seeking therapy early enough.

"*So that people can clearly know the actual cause of the death of a person, not that they are left to imagine.*"

**FGD 009 Respondent 2**

"*The results of the DNA should be given to me because it was part of my body that was removed, it was nobody's body part, it was mine.*"

FGD 007 Respondent 10

"*I think the results should be given to you because by the time you went for the test you wanted to know your health status so the results should be given to you so that you can know about your health status better.*"

**FGD 007 Respondent 2**

**1.1 Therapeutic and/or diagnostic misconception.** Respondents highlighted several benefits associated with provision of feedback of genetic testing results including the fact that individuals are helped to plan their lives and the wellbeing of their families. They felt that the results could guide medical professionals and scientists to search for treatment, and institute preventive measures before a disease manifests. Others felt that it will be an added advantage because they will have gained more health information about themselves to help predict the future.

"*I think it is basically the knowledge after getting the information that really prepares you to be free. Now like us at least we have heard and we have gotten to know what it is all about, so it gives me the freedom (courage) to participate freely without the fear that I had before.*"

**FGD 006 Respondent 4**

''*It would help me know what the illness is and whether the complication is from my mother or my father, so that I can alert them and see how to protect my children.*''

FGD 009 Respondent 1

''*When I receive feedback at the right time and there are no other discouragements and at the same time the person who is giving me feedback first begins by counselling and guiding me, reminding me of what went on and how to live afterwards.*''

**FGD 006 Respondent 1**

Others noted that genetic testing and associated feedback of results is good because they get to know their diagnosis and can plan to prevent future illness. Some thought it an added advantage because in certain circumstances, individuals live with uncertainty and suspicion concerning particular traits that could be eliminated through testing. A desire for ancestry testing was expressed.

*"We know that these diseases could have come from the ancestral line of our parents, so knowing the result is good because you can be able to trace whether it is coming from your mother's line or father's line and inform them to protect the next generation of the family.'*

**FGD 009 Respondent 6**

*"Yes, I want to know the results of this DNA test because it helps me to know the status of my blood and also know my clan too.*"

**FGD 008 Respondent 1**

**1.2 Concerns, Challenges and implications for sharing.**   Respondents noted that although GGR and the associated feedback of results are important, they were worried about the cost of such tests. They felt it might be too costly for them in case there was need for such genomics testing post the research period. Hence respondents appealed for affordable genetics testing within reach of the low-income earners. There was also concern about being diagnosed with a condition where the cost of treatment is prohibitive.

''*The problem is, it is a bit expensive and then those services are very far, otherwise I would really say that it is a good thing to do.*''

**FGD 006 Respondent 1**

**2. Paternity information as a benefit and risk.**   About one third of the respondents including both men and women linked participation in GGR and receiving feedback of results to establishing the paternity of their children and of other family members. A number of respondents had experience in using DNA related technologies to solve social issues.

*"I want to acknowledge that my father made my sister to go through the same. At first, he denied being the father to my sister but when they went for a DNA test, it was confirmed that he was the true father. He no longer has any doubts and he is instead happy now.*"

*FGD 006 Respondent 7*

*"In fact, this has happened to me before; my husband denied my second child saying I cheated and when we went to the hospital to prove, their DNA was the same and he even did not apologize for accusing me of adultery".*

**FGD 009 Respondent 4**

''*It is a good thing because there have been cases of domestic violence because of a man doubting the paternity of some of his children, such would help solve some of these problems causing violence in the homes.*"

**FGD 009 Respondent 6**

Some respondents noted that feedback of GGR results has the potential to reveal discordance in paternity in case both the child and male parents are tested, and this has the potential to cause family disruption and associated psychological harm and suffering of the individuals involved.

"*The DNA is what distinguishes one person from another. For example, you can tell that this child does not belong to this family and does not belong to the other family. . . It means that, that child should not stay with that family and the mother should take that child where it rightfully belongs.*"

**FGD 006 Respondent 7**

"*With results there are two things involved. If the results turn out to be good, this brings happiness, but if the results are not good, this will automatically bring violence in the house.*"

**FGD 008 Respondent 9**

## 3. Considerations for feedback of findings

Although respondents expressed willingness to receive feedback of genetics testing results, they highlighted several requirements that need to be put in place before results are shared.

**3.1 Adequate Informed consent.**   Respondents observed the need for adequate informed consent before the tests and at the time of feedback of results. Informed consent would facilitate individuals' understanding of what they are getting involved in, highlight the associated implications and aid informed decision making for GGR participation and the feedback of results. As highlighted below

"*May be improving on learning resources or materials to make people understand things better.*"

**FGD 006 Respondent 6**

"*Availing information out there to the people explaining the importance of doing it and why we do it can really play a very big role in causing (a positive) change in attitude and perception of the people. So that is what I would suggest that we continue doing. It will be good to reach out to more people.*"

**FGD 006 Respondent 7**

''*If they come and teach me very well on what exactly they want to do plus let me know of the cons I can accept to participate in this DNA testing, it will also let me know my health status and they will also help me in case I have any health complications.*"

**FGD 007 Respondent 3**

''*If they teach me well and I fully understand how this research works I can accept to participate.*"

**FGD 007 Respondent 8**

''*If they also tell me very well whereby, I also fully understand this research I can accept to participate so that I can be evidence to the community to let them know that it is not a bad initiative after all so that the research runs smoothly.*"

**FGD 007 Respondent 5**

Respondents highlighted the need for research teams to facilitate participants' understanding of the genetic information through a process which would include the use of visual aids in order to facilitate the information delivery process and promote understanding.

''*What I think is that there should be a projector to show us photographic images of this genetic science and the procedure of genetic testing. When you see that the other child resembles the parents it makes you to appreciate that you are studying something that exists.*"

**FGD 006 Respondent 5**

'*If the research department could show a video, somebody who does not know how to read will be able to interpret what is going on and gain interest in participating in testing and receipt of results.*"

**FGD 006 Respondent 4**

Respondents highlighted the need for a clear appreciation of the genetic condition likely to be identified by the test results so that they are well informed of what is likely to happen to their bodies. They stressed the desire to know the results to the extent possible and if the information turns out to be complicated, then the provision of feedback should involve their parents or close relatives. Hence, the need for the informed consent process to employ visual aids to facilitate participant's understanding.

"*If it were possible, I would want to see the nature of the disease through an image or explained to me thoroughly.*"

**FGD 009 Respondent 3**

"*I will accept because they would have taught me and I would have understood very well what they want to do. This will allow me to make up my mind and also I will know exactly what to do and also know what exactly is needed for my life.*"

**FGD 007 Respondent 3**

**3.2 Genetic counselling.**   Adequate genetic counselling by a well-trained professional, preferably a doctor was considered essential for participants before getting their genetic testing results. Respondents felt that good counselling would help allay anxiety associated with receiving genetics results. Counselling would also help address any misconceptions or misunderstandings associated with genetics and genomics.

Many respondents preferred to receive the feedback of results in person. Respondents felt that this would give the person providing the feedback an opportunity to guide them on health facilities that could provide the requisite medication.

"*The counselling that is given before feedback of results from there (research centre) can encourage me and even my clan members to do that test.*"

**FGD 006 Respondent 3**

"*Of course, as doctors they know how to describe the results of a test; they should be professional in breaking the news to me.*"

**FGD 009 Respondent 2**

"*Whoever is breaking the news has to sit calmly face to face with me to explain the results for me to understand well, without sending me into a shock.*"

**FGD 010 Respondent 4**

Respondents felt that prior to feedback of results and the associated health information, there was a need for education on any available options for managing the resultant health disorder. For example, if there was a possibility of managing the health condition, this should be stated before the condition is disclosed. They also observed that it is important that enough time should be spent with the participants when disclosing findings.

"*The doctor should first counsel me because sometimes if they find a disease and they just give me a paper, it can make me unsettled but if the doctor talks to me, tells me that we found a disease but take care of yourself, take your medication, this will eradicate my fears.*"

**FGD 008 Respondent 10**

"*It is not right for a doctor to give my results to someone else to bring to me. This is because that person doesn't have the experience equivalent to doctors.*"

**FGD 007 Respondent 4**

"*I desire to be counselled properly before giving me the results.*"

**FGD 010 Respondent 6**

"*If my samples were collected from home, it means that my results should also be brought home. They should not abruptly return the results. before giving me results, they should counsel me. My mind should be settled before giving me my results.*"

**FGD 007 Respondent 8**

**3.3 Privacy and confidentiality.** Respondents stressed the need for a quiet and private environment at the time of disclosing the results. Since genetics testing results is regarded as private information, the need to observe privacy and confidentiality is a growing reality that should be respected at all times. Many respondents proposed that at the time of disclosing findings, it should be only the doctor and the individual who was tested.

"*Earlier you talked about confidentiality which automatically means that in case they pick my sample for DNA my results will definitely be given to me meaning that whatever it is it will be between me, the doctors and the people carrying out the tests.*"

**FGD 007 Respondent 10**

"*The results should be given to me directly. In case they find any medical complications, the person who has brought the results should explain to me their findings and also if possible, bring medicine and prescribe for me how to take the medication. If you give the results to someone else the person will begin telling people behind my back how my condition is very worrying and bad.*"

**FGD 007 Respondent 7**

While most respondents felt that the results should be shared directly with the individual who was tested, some thought that they would need support and presence of a family member at the time of getting the results. Others proposed that if it's a condition that affects the family, then the doctor can disclose to the whole family. This would help everyone to know about the condition in the family.

"*For me I want to be with my parents when I am getting the results from my hereditary testing.*"

**FGD 010 Respondent 5**

"*The results should be given to me personally because it is me supposed to tell my parents and also, I would want it in written form because the records can help me in future, let's* say *my condition becomes worse, I need to show those results at the hospital. If you get your results via message, you can't go and show a message to a doctor so I feel it's better to receive in written form.*"

**FGD 007 Respondent 5**

## 4. Extending feedback of genetics findings to family and community

**4.1 Extending feedback to family.**   Many respondents expressed the wish to share their genetic results with some family members but this can only be done after adequate consent and genetic counselling to evaluate the possible implications of such sharing. Many families in Uganda share health and medical information because to a great extent the family members meet the cost of treatment. Sharing genetic results is no exception particularly if the genetic predisposition has health implications. Additionally, genetic predisposition can affect other members of the family, and such sharing would act both as a warning and a preventive measure. It would also facilitate future planning for the concerned family. Respondents had varied opinions on extending feedback of results to their relatives with some stressing that the results belong to the individual who was tested, while others thought they could share findings with close family members. Some highlighted the fact that if one is likely to suffer from a genetic condition, then it was necessary to share the findings of genetic testing with individuals who will take care of them in case they become sick.

"*The family members need to know because some diseases may need extra attention and care like meals on time, special foods etc, so that the family members can be helpful in looking after you.*"

**FGD 010 Respondent 5**

"*I also feel it is right to tell my parents because it gives my brothers and also my wife the opportunity to also go and test in case, I turn out to be positive of any illness so that other children don't inherit the diseases too.*"

**FGD 007 Respondent 8**

Other reasons for extending feedback of genetics results to family included the need to inform others and help them know of their predisposition to disease early so as to take appropriate action. To others, genetic information was considered a family health issue which affects all members of the family and so they have to be told the results.

"*To me I think it depends on the type of disease because sometimes it might be a non-life-threatening condition or it can be like epilepsy which doesn't go hand in hand with loud noise (which is thought to trigger some epileptic attacks) so the people back home should know how to handle me. So, if the feedback of results is to me, then at least my parents have to be there and also, they should put in the records so that in case another disease comes in, the doctors will have an idea on how to help me.*"

*FGD 007 Respondent 8*

"*It helps the family to understand the problem that they are faced with so that it's able to plan together, how to help in case there is any one sick and others are not, the family can understand how to plan and handle such situations.*'

*FGD 006 Respondent 1*

"*Because it helps on the side of treatment and health and unity of the family.*"

*FGD 006 Respondent 8*

Reasons for not extending feedback of genetics testing results to family included the feeling that such information is individual and private, the fear that some members may not understand the meaning of such information or their inability to handle the associated stress and anxiety.

"*My view is that your test results should only be given to you because they are private and will only affect you.*"

**FGD 010 Respondent 1**

"*These results will remain in my house; I will not share them out anyhow.*"

**FGD 008 Respondent 5**

**4.2 Extending feedback of genetic results to the community members.**   Individuals in the Ugandan rural settings live as communities, often sharing health information and assisting in care of others including contributions to the treatment costs for those in need. If genetics results point to a particular predisposition that may affect the health of an individual, their family or community, such information may be shared by the individual tested at their discretion with clear appreciation of the implications. Some respondents felt it was acceptable to share their genetics results with the community particularly those they thought would support them in case they became unwell. Others felt that DNA test results do not necessarily mean disease. Hence trusted people who may not be relatives, can know the results and advise.

"*For me I would tell all of them, so that they are aware of what the doctor has advised me to do and stand with me in support.*"

**FGD 010 Respondent 3**

"*It is good to share results with other people because it safeguards their health. If a person knows that I have a particular genetic disease, it will be up to them to decide whether to produce with me (children) or not. If a person chooses to marry me, that is their risk.*"

*FGD 006 Respondent 7*

"*It is good for the community to know because these days people assume someone's death up to the extent of accusing other community members with whom the person could have had a grudge, to avoid such assumptions, they should know.*"

*FGD 010 Respondent 2*

"*I will accept that my wife should know, the community should also know, there are other diseases that can be spread, those near me can even help me if I am weak, my neighbours should also know.*"

*FGD 008 Respondent 1*

Respondents who did not favour extending feedback of genetics results to the community thought that this was private health information for the family that should not be shared with non- family members. Others felt like sharing such information exposes the health condition of the family and that might end up causing the family to be ridiculed or stigmatised.

"*Yes, my results are important to my family members to know but not outside of family because sometimes that is a secret we have in our house.*"

**FGD 008 Respondent 10**

"*It depends on the type of disease, if it is a disease that I can survive with by taking care of myself I feel it is ok to keep it to myself but if it is a condition that needs people's help like me getting lost, then the community should know about it.*"

**FGD 007 Respondent 8**

"*I think it's a bad idea because people who do not like you take advantage of the information to spread bad information about you and you become the talk of the town, so I think it's best to give it to the owner of the results.*"

**FGD 010 Respondent 5**

**4.3 Strategies for sharing feedback of genetics and genomics research results.**   Regarding the strategies for sharing feedback with family, several approaches were suggested by the respondents. Some respondents felt that they should have exclusive rights to disclose the information, hence the doctor should provide them with enough information that can be used to inform others. Some respondents felt that the doctor is better placed to provide feedback of results to family members because they are assumed to be better informed and equipped with the necessary genetics counselling skills. While others thought that they would pass information to an elder in the clan or family who would in turn take the responsibility of conveying the information to the rest of the family through approaches like family meetings.

"*Alternatively, the testing team can come home pick samples one by one (testing can be carried out at the home of the participants), it will be right to counsel me together with my parents so that they can know what to do, those are the ways our results can come back to us.*"

**FGD 007 Respondent 6**

## Discussion

We set out to assess the views of grassroots communities in Uganda on if and how feedback of hypothetical genetics results should occur and if they were willing to participate in GGR. Our study findings show that such feedback of results was acceptable to all respondents conditional upon receiving health status information. Several reasons for needing feedback of results were identified including and especially, the need to know one's health status and planning for the future. Several strategies were proposed if such feedback was to be conducted appropriately.

Receiving genetics results that impact on one's health condition can be a benefit to research participants particularly in an African rural setting where genetic testing is out of reach of almost all individuals and access to general public health services is limited. Feedback to research participants is a growing reality and an ethical obligation that should be incorporated in the research processes as highlighted by several ethics guidelines [15, 29, 30]. Although the usual call for feedback of research findings has been focussed on other fields of research, the need for return of results is emerging in genetics research because of the anticipated implications such as fear and anxiety related to uncertainty around impact of findings on future health or other family members.

The fears associated with findings of genetics research can be appropriately addressed via mechanisms like adequate consent processes and genetic counselling. However, qualified genetic counsellors are an extremely limited resource in Uganda and in most African countries [26]. Other mechanisms to address the fears include observance of privacy and confidentiality as well as only sharing results that are potentially beneficial or actionable. Related work among genomics research participants and genomics researchers in Uganda has also highlighted the need for feedback of GGR results [25, 26].

Although researchers in Uganda are supportive of return of individual genetics results, they were hesitant to share results due to complexity of the science and lack of staff to accurately communicate the meaning of results, the lack of context specific guidelines as well as the challenges of accurate interpretation of the findings [26]. This dilemma faced by researchers is worsened by the desire of grassroots communities to receive feedback of all their GGR findings with clear interpretation whether beneficial/actionable or not. Participants in this study conflated genetics and genomics results even though it was explained that only significant and actionable genetics results could be returned in the information session prior to the FGDs. The need for feedback of genetics results has been described as a form of solidarity by research participants in Botswana and as a reciprocity obligation of researchers who can make participants feel valued as part of a mutual relationship [19]. Dissemination, beneficence and reciprocity have been considered as essential components of a framework for enhancing ethical genetics research with indigenous communities in the USA [31]. It should be noted that, the hypothetical research scenario the participants are responding to in this study may not correspond to the reality of feedback, and this should be borne in mind when extrapolating on the basis of these study findings.

Participants felt that knowing the outcome of GGR testing would help them seek early treatment or prevention, creating the impression that genetic testing results were reliably associated with causality of disease and that treatment is available for all conditions caused by a genetic predisposition. Likewise, there was an expectation that genomics results would confer

individual benefit. Although this was a rural non-research setting, it is important to note that therapeutic and diagnostic misconception where participants perceive research as diagnosis or care rather than experimentation are very common [32–40]. Such misconception may lead individuals to participate in research with diagnostic or therapeutic expectations as has been observed among tuberculosis genetic study participants in Cameroon [32]. Yet most genetic studies may not yield results that can benefit health or predict risk of disease [41]. But even in cases where accurate diagnosis can be made, many diseases identified may not be treatable.

For feedback of genetics results to be conducted appropriately, several strategies were proposed by the respondents including adequate consent processes, genetic counselling as well as privacy and confidentiality. Informed consent for research participation is an ethical requirement that should be carried out as a continuous process starting before recruitment, through implementation to the post study period. Such consent processes should be suitable for participants and be provided in a language that is easily understandable to the participant. And for feedback of genetics findings, consent should be done at different levels including before recruitment of the participant, and at return of results for those who chose to receive results. The need for meaningful informed consent has been highlighted by participants of a genomics research study in Uganda [25]. However, the referenced study also revealed participants' recall bias about their participation in the concerned study that affected their shared experience on the informed consent process. GGR has been challenged by the fact that genetics and genomics terminology and associated vocabulary may be difficult to translate into many of the local languages in Uganda. Hence making it difficult to achieve adequate and meaningful consent.

Recent work that reviewed consent documents for 13 H3Africa genomics projects observed that genetics was mostly explained in terms of inherited characteristics, heredity and health, genes and disease causation, or disease susceptibility and only one project made provisions for the feedback of individual genetic results [42]. However, it is important to note that not all GGR can return individual results for example genetic epidemiology of complex disorders or population genetics results which focus on the population overall. Challenges regarding meaningful informed consent for GGR have been observed particularly when it comes to sharing of human biological samples and data in the context of international collaborative research [43, 44]. In order to address the challenges associated with informed consent in GGR, some commentators have proposed tailoring the informed consent process based on a ten-point framework. The framework includes among others the study design, data and biological sample sharing, reporting study results to participants, cultural context, language and literacy and potential for stigmatization of study populations [45]. However, this proposed framework needs to be clearly interpreted and studied if it is to be meaningfully applied.

In additional, for consent to be meaningful it should be coupled with relevant information on the proposed genetics testing and associated implication. Such genetic counselling is essential and should be provided before testing and during feedback of results. It should be noted that although genetic counselling is a developing field in emerging economies like South Africa [46], there is a relative lack of qualified genetic counsellors and the associated counselling in many of the low resource settings in Africa including Uganda [26, 47]. Yet such genetic counselling would go a long way in addressing issues like implications of GGR and feedback of results to the individual, the family and the community. Other issues that can be addressed by counselling include aspects like therapeutic misconception, privacy and confidentiality as well as the common beliefs in the Ugandan setting of genetic testing being associated primarily with paternity testing. Since the concept of genetic counselling is relatively new in our setting and virtually non-existent in the rural communities, respondents felt that the doctor who is most knowledgeable should be the one to conduct the counselling. The lack of genetic counsellors can be addressed by capacity building for genetic counselling.

Furthermore, consent forms should be explicit on aspects like who would have access to genetic results and whether return of results concerning paternity information would be done. If so, this should be approved by the REC before data collection is initiated. Otherwise, it is always a dilemma when researchers discover sensitive information after running the tests and seek guidance from an equally unprepared REC. For example, during genetic testing for sickle cell disease, which is prevalent in Uganda, it's not uncommon to discover discordant genetic information between the child and the male parent. Similar findings emerged in a study in Kenya that discussed challenges associated with paternity mal-alignment [48]. It would be good if consent documents provided by researchers and approved by RECs clearly state if such paternity information including misaligned paternity will be provided to both parents.

Our study findings highlight a situation where participants stress the need for privacy and confidentiality of their participation in GGR and return of results. However, despite the call for privacy and confidentiality, most of the respondents preferred the presence of a family member during feedback of results, hinting that such feedback could as well be done at participants' homes. Hence the concept of confidentiality in these communities needs to be clarified and could imply keeping information not only to the individual tested but within their close family. The nature of confidentiality proposed by our respondents is quite different from the model practiced in the western world where private information usually remains with the individual tested. Additionally, the privacy and confidentiality mandated in the research ethics guidelines and some laws like the Data Protection and Privacy Act of 2019 in Uganda [29, 30, 49], employ the western approach that focuses on individual privacy and not the family as a whole. Hence the need for adaptation to fit the local setting. However, it should be noted that anonymisation of genomics data may not always be possible.

Other considerations to facilitate understanding of the GGR concepts include meaningful community engagement (CE). Such engagement would help researchers understand community-based practices for example the meaning of privacy and confidentiality, and whether it should be handled at the individual level or family level. Some commentators have proposed the Tygerberg Research Ubuntu-Inspired Community Engagement Model. This model would require RECs/IRBs to play a role in requiring a CE plan for every study that is community based, and scientific journals to require a paragraph on CE in publication of relevant research projects. This would ensure moving CE from a guidance requirement to a regulatory requirement, emphasizing that it is a critical component of a robust consent process in research and that it ought to be embedded within research projects, where applicable [50]. However, for such community engagement to be effective in facilitating research that is responsive to community needs, it has to be meaningful and collaborative. Such collaboration may extend to levels like community involvement, community empowerment and community participation. The collaboration can facilitate applicable benefit sharing models, developing capacity for genetic testing, counselling, technology transfer and translation of vocabulary.

Many respondents were agreeable to extending feedback of genetics testing results to family because genetics information was considered to belong to the whole family since it is inherited. The need for extending feedback to family and sometimes the wider community could be explained by the fact that most of the individuals in the Ugandan rural setting live in communities, often share health information and support each other during times of sickness. It is also important because family and community members play an important role in the provision of health care to patients. However, despite the fact that most of the respondents were agreeable to extending feedback of GGR results to family, fewer respondents supported such feedback to the wider community and only for particular health conditions. Sharing of genetics results with family and community was assumed to have a potential to benefit the individual tested in terms of social and financial support in case the individual became unwell, but

would also act as an early warning to others at risk. Extending feedback should depend on the willingness of the individual tested, nature of the genetic condition, adequate genetic counselling and appreciation of the implications and, if there is likelihood of the tested individual to benefit from such disclosure.

The perception regarding extending feedback of genetics research results needs to be studied further and should be done on a case-by-case basis because the implications may vary amongst individual patients and from one family to the next. If genetics research results point to a particular predisposition that may affect the health of an individual, their family or community, extending feedback to family and community may be done at the discretion of the individual tested based on a clear appreciation of the implications. And such sharing may involve general information not specific personal information. This is in line with recommendations from a USA consultative team involving an ELSI working group of national experts which, among other recommendations, suggested that researchers should elicit participants' preferences on such extension of feedback to family but also recommended further research on the subject matter [51]. Other countries like Australia, Britain and France have legislation that allows healthcare professionals to disclose genetic information considered beneficial to family members in case the individual tested is not willing to do so [52–54]. While current Belgian legislation coupled, with international precedent, may provide sufficient justification to establish a duty to inform relatives of their genetic risk in some cases [55].

It is imperative that the privacy and confidentiality of the person enrolling in the study be respected. But in cases where there is benefit in sharing results with family members, the original participant should grant permission because just like feedback to individuals, feedback should not be imposed on family members, but should be based on their voluntary consent [14]. A study involving REC chairpersons in the USA showed that 62% of the REC chairs agreed that participants should be informed that their results could be offered to family members and asked to indicate their choice, but such a statement may not be adequate informed consent [56]. Keeping genetic information and associated diseases confidential may be very difficult particularly in the Ugandan setting where the costs of medical treatment to a great extent are met by the relatives and sometimes the wider community who may inadvertently learn of the patients' genetic condition. Since individuals in the communities are agreeable to extending feedback of GGR results to family members, it's up to the research regulators with the oversight mandate to devise appropriate frameworks and guidelines to guide the process while respecting the participants' preferences, privacy and confidentiality.

Although feedback of results is generally acceptable by grassroots communities, it is necessary to adequately evaluate and address the implications associated with such testing and feedback of results. Implications may include psychological and socio-economic aspects such as family disruption, a feeling of low self-esteem, stigma and discrimination, denial of insurance or increased premiums. Such effects may affect the individual tested, their family or the wider community. Yet familial implications of genomics contribute to the need for extending feedback to family. Hence adequate genetic counselling both before testing and at feedback of results, appropriate consent as well as maintenance of privacy and confidentiality are necessary for GGR.

Other challenges associated with return of GGR results include lack of appreciation of the information due to unfamiliarity with genetics and genomics vocabulary as well as affordability of the testing services to the community post the research period. Hence the need for appropriate translation of the information to a language understandable by the participant, use of visual aids to support information delivery as well as use of non-technical terminology. Additionally, the lack of availability or affordability of genetic testing facilities to the community post study, lack of a clear interpretation of the results and meaning of the findings may

cause confusion. The researchers should therefore build meaningful collaboration with communities where research is conducted, and share only confirmed results and information with participants. This will avoid the negative implications of such research and enable research communities to benefit from the results of research.

Finally, most GGR conducted in Uganda to address ethical, social and legal issues has been carried out in well-established research settings, and the views that have informed debate on ethical conduct of GGR in the country are mainly those of research participants, researchers and research regulators [23–26, 57]. Our addition of the grassroots communities will contribute a new dimension with an additional group of stakeholders whose views will enrich the literature as well as the targeted ethics guidelines for conduct of GGR in Uganda. The acceptability of grassroots communities to participated in GGR as well as receive feedback of results whether beneficial/actionable or not goes beyond the protective recommendation by researchers and research regulators of sharing only beneficial/actionable findings. We believe the ethics guidelines for conduct of GGR informed by our study findings will go a long way in informing regulation and oversight of GGR in the country.

## Limitations of the study

The individuals who participated in the study were research naive and may not have fully appreciated the implications of participation in GGR and feedback of the associated results. To address this aspect, a deliberative approach to FGDs was used and included an educational component to help the respondents appreciate GGR. However, the depth of understanding required for GGR may not have been achieved in 30 minutes and a longer period of engagement with similar communities should precede GGR studies. In particular, the explanation of the difference between genetics research and genomics research probably required more time and use of additional educational tools.

Since the study was conducted in three different languages, the researchers needed assistance from individuals fluent in the respective languages to conduct the FGDs and this might have affected the quality of the interviews and the subsequent data.

Additionally, the FGD were conducted in three different local languages and later translated into English which could affect the quality of data. These challenges were mitigated by identifying research assistants with good experience in qualitative data collection, protocol training of the research assistants to understand the study and using well translated data collection tools.

Given the fact that this was a qualitative study, although the findings provide a deep understanding of the subject matter, they may not be generalizable. However, a wider range of other stakeholders have been involved in related research which enriches the generated data. Stakeholders like researchers and REC members have been involved in in-depth interviews and quantitative surveys, genomic research participants and patients have been involved in FGDs.

## Conclusion

Participation in hypothetical GGR as well as feedback of genetic testing results is acceptable to individuals in grassroots communities conditional upon receiving health status information, to establish paternity in some cases and to plan for the future. The strong diagnostic and therapeutic misconception linked to GGR is concerning and has significant implications for consent processes, community engagement, genetic counselling and research ethics guidance. Furthermore, the expectation of paternity testing results being embedded in all GGR needs to be managed appropriately. Given the misperceptions and unrealistic expectations expressed, it is an ethical imperative to build meaningful collaboration with research communities for appropriate genomic language/vocabulary translation, benefit sharing, capacity development

and knowledge translation. Finally, the willingness of grassroots communities to participate in GGR as well as receive feedback of genetics results whether beneficial/actionable or not goes beyond the protective recommendation by researchers and research regulators of sharing only beneficial/actionable findings and should be further evaluated. While extending feedback of genetic research results to close family members was generally acceptable, concern was expressed in extending feedback to the community.

## Supporting information

**S1 File.**
(DOCX)

## Acknowledgments

We are grateful to all the individuals in the various communities who participated in this study.

## Author Contributions

**Conceptualization:** Joseph Ochieng, Betty Kwagala, Marlo Möller, Keymanthri Moodley.

**Data curation:** Joseph Ochieng, Betty Kwagala, John Barugahare.

**Formal analysis:** Joseph Ochieng, Betty Kwagala.

**Funding acquisition:** Joseph Ochieng, Betty Kwagala, John Barugahare, Keymanthri Moodley.

**Investigation:** Joseph Ochieng, John Barugahare.

**Methodology:** Joseph Ochieng, Betty Kwagala.

**Project administration:** Joseph Ochieng.

**Resources:** Joseph Ochieng, Keymanthri Moodley.

**Supervision:** Marlo Möller, Keymanthri Moodley.

**Validation:** Joseph Ochieng.

**Visualization:** Joseph Ochieng.

**Writing – original draft:** Joseph Ochieng.

**Writing – review & editing:** Joseph Ochieng, Betty Kwagala, John Barugahare, Marlo Möller, Keymanthri Moodley.

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
