## [Decision Letter · Decision Letter 0]

28 Jun 2022

PONE-D-22-10220Feedback of Individual Genetic and Genomics Research Results: A Qualitative Study Involving Grassroots Communities in UgandaPLOS ONE

Dear Dr. Ochieng,

Thank you for submitting your manuscript to PLOS ONE. After careful consideration, we feel that it has merit but does not fully meet PLOS ONE’s publication criteria as it currently stands. Therefore, we invite you to submit a revised version of the manuscript that addresses the points raised during the review process. Please submit your revised manuscript by Aug 12 2022 11:59PM. If you will need more time than this to complete your revisions, please reply to this message or contact the journal office at plosone@plos.org. Please include the following items when submitting your revised manuscript:A rebuttal letter that responds to each point raised by the academic editor and reviewer(s). You should upload this letter as a separate file labeled 'Response to Reviewers'.A marked-up copy of your manuscript that highlights changes made to the original version. You should upload this as a separate file labeled 'Revised Manuscript with Track Changes'.An unmarked version of your revised paper without tracked changes. You should upload this as a separate file labeled 'Manuscript'.

We look forward to receiving your revised manuscript.

Kind regards,

Angeliki Kerasidou

Academic Editor

PLOS ONE

Journal Requirements:

2. In the ethics statement in the Methods and online submission information, please ensure that you have specified what type you obtained (for instance, written or verbal, and if verbal, how it was documented and witnessed). If your study included minors, state whether you obtained consent from parents or guardians. If the need for consent was waived by the ethics committee, please include this information.

“Research reported in this publication was supported by the National Human Genome Research Institute of the National Institutes of Health under Award Number U01HG009822 and NIH Fogarty grant under Award Number D43TW01511. The content is solely the responsibility of the authors and does not necessarily represent the official views of the National Institutes of Health.”

Reviewers' comments:

Reviewer's Responses to Questions

**Comments to the Author**

1. Is the manuscript technically sound, and do the data support the conclusions?

Reviewer #1: Yes

Reviewer #2: Partly

Reviewer #3: Yes

2. Has the statistical analysis been performed appropriately and rigorously? 

Reviewer #1: N/A

Reviewer #2: I Don't Know

Reviewer #3: N/A

3. Have the authors made all data underlying the findings in their manuscript fully available?

Reviewer #1: Yes

Reviewer #2: Yes

Reviewer #3: No

4. Is the manuscript presented in an intelligible fashion and written in standard English?

Reviewer #1: Yes

Reviewer #2: Yes

Reviewer #3: Yes

5. Review Comments to the Author

Reviewer #1: This manuscript reports on the results of deliberative focus groups conducted with members of grassroots communities in Uganda. These focus groups explored views on return of results from hypothetical genomic research. The findings are interesting and important due to the paucity of research on this topic in non-USA populations. As a general comment, the manuscript would benefit from proofreading for punctuation. I have the following specific comments:

Abstract

Line 58. Please specify which type of content analysis was employed (ie., deductive versus inductive).

Line 69. Full stop missing.

Line 70. Use GGE rather than writing in full.

Line 72. The authors state “The strong therapeutic misconception linked to GGR is concerning…” but this doesn’t make sense unless the results within the abstract mention that the hopes for benefit are unfounded.

Introduction

81. Really long sentence. Consider splitting up. It would also be good to explain these challenges in a little more detail.

94. Consider deleting “with resultant literature” as it is implied.

95. Another really long and hard to follow sentence.

104. The authors state debate is limited but then list 7 studies. It would be good to highlight what exactly about these papers is lacking to justify the need for the current study.

113. It would be good to define grassroots communities.

Methods

128. I think the authors mean focus groups, not interviews.

131. The information about the number of focus groups conducted should be in the results section.

139. Please provide more information about how participants were recruited. How were the selected? Who approached them and how?

161. Please provide a reference for the type of analysis you used.

Results

It’s great that the authors have listed which FG the participant quotes are from but it would also be good if each participant in each FG was numbered as well. This adds rigour to the results because it shows it’s not the same person speaking all the time, eg. lines 358 -366.

189. Content analysis identifies content categories, rather than themes.

215. Can the authors please clarify whether positive and negative here mean gene-positive/negative or positive/negative from the point of view of the participant? Not currently clear.

446. This quote was already used in the genetic counselling section. Not great form to reuse quotes.

448. Use of the word ‘noise’ doesn’t make sense to me. Maybe check translation?

Discussion

557. Are these suggestions for addressing fears based on your data or other literature?

590. Again, I don’t think you really made this point about the therapeutic misconception explicit in the results. Make it clearer so we know what you are referring to when you discuss it in the results.

Consider breaking up some of the long paragraphs into separate ones for ease of reading.

681. Is this true? Australian laws allow this but I am not sure about Belgium so I would check this. There’s a paper by Phillips et al from last year that specifically discusses Belgian Laws on this issue.

It would be good if the findings of this study were contrasted more with others, particularly from rural communities.

Reviewer #2: The aim of this paper is to assess the perceptions of grassroots communities in Uganda regarding the feedback of genetic and genomic results to patients/participants, presumably with the goal of helping to inform guidelines and or policy regarding the feedback of individual genetic/genomic results in this context. I have a few questions regarding the author’s interpretation of their results and statement of their conclusions, which would need to be addressed prior to the paper being suitable for publication.

1) The primary issue is with the author’s claim that, according to their results, the feedback of genetic/genomic results “was acceptable to all respondents” (Line 566). First, it would be worth being explicit that this acceptance comes with certain conditions, which the authors describe in the body of the paper (e.g., informed consent, the condition being treatable). More importantly, however, it is not clear that what the feedback of research results would involve in practice is relevantly similar to what the participants in this study envision, and thus, I’m not sure that the authors can really say that these participants would find the feedback of genetic/genomic results ‘acceptable.’ The authors tacitly acknowledge this when they raise the issue of the therapeutic misconception, and point out that most genetic studies may not yield results that can benefit the participants. Yet this seems to be a pervasive assumption on the part of the participants. I think the authors need to be clearer that the hypothetical situation the participants seem to take themselves to be responding to may not correspond to the reality of feedback, and this should be borne in mind when extrapolating on the basis of their study findings.

2) Similarly, it would be useful if the authors could be a bit more specific about the context in which feedback is occurring. Is it research involving genetic/genomic testing? Genetic/genomic testing as a part of clinical care? Are the results only individual-level research results, or might this include secondary or incidental findings?

A few other small points:

3) On Line 87, the authors mention ownership of DNA; does this mean ownership of the biological sample, or the data derived from sequencing the genome?

4) On Line 101, the authors state that there is consensus that several conditions must be met before feedback of results is permissible, including “ensuring that patients are appropriately informed of the implications of the findings.” If the question is whether feedback should happen or not, how can informing the patient of the implications be a deciding factor (in order to inform them of the implications, wouldn’t we needed to have informed them already?)

5) I wasn’t sure what the authors meant when they stated that coding was done ‘both inductively and deductively’. A bit more description of how new codes were generated, and their justification, would also be useful.

6) It wasn’t clear to me exactly why ‘grassroots communities’ in particular were an important population to engage with on this issue; is there a reason to think that their views would be different than participants in other contexts. More generally, a few sentences early on describing why the numerous ethical guidelines for feeding back results might not be applicable to the Ugandan context would really help to motivate the paper. The section beginning on Line 647 is a really nice example of why policies developed based on other socio-cultural contexts might not be transferable to the Ugandan context, and it would be good to highlight this as one of the important

Reviewer #3: This paper reports the findings of a series of deliberative focus group discussions exploring community attitudes to the feedback of genomic research results. It provides a valuable addition to the literature by reporting on Ugandan community attitudes, a region relatively new to genomic research and with whom little research of this nature (ELSI) has been undertaken. The findings have the potential to usefully inform guideline and policy development in this region.

However, there were some problems with the manuscript that require attention prior to publication.

Major:

* 'research results' needs to be clearly defined early on. It's really not clear what you mean by this (overall research findings? individual health information? raw genomic data or interpreted info? just actionable results, or any findings of interest?) and this has implications for interpreting the results. There is no reason why individual health info should be shared or would be shared with the whole community, but it is widely expected that research findings will be communicated with the wider participating community, especially if there are potential ramifications for socially identifiable groups. It's also not clear whether this was adequately explained to the focus groups in the education component.

* In the introduction, you need to briefly outline the findings of previous research, rather than just say that 'There have been a few publications on perspectives...' - this will give readers an idea of what to expect, and is also an opportunity to highlight why we might expect Uganda/grassroots communities to differ from other cohorts.

* on what basis were the communities selected?

* what language were focus groups held in? transcripts? were you interpreting as you went, or were discussions translated later? were back translations employed for accuracy checks?

* what do you mean by 'a safe distance' was this about Covid or people overhearing?

* the qualitative analysis sounds more like a form of thematic analysis than content analysis? the methods section would be strengthened by references to relevant theoretical/methodological literature and greater detail on what was done

* how well was the research context and how it differs from the clinical context discussed/understood? there is significant evidence of therapeutic misconception throughout - some of this just be how people understand research, but some may have arisen through lack of clarity from the team here, eg the participant refers to themselves as 'the patient', they all want to know their 'health status' and assume they are sick or the researchers will find a disease

* actually, 'therapeutic misconception' would be a useful thematic section in the results

* the results section requires significant reworking - there are numerous sections where the illustrative quotations don't relate to the point just made, and other sections where points are out of order. The quotations chosen have a lot of potential to be illustrative of interesting and useful points, but are currently not well integrated.

* in a lot of places, phrases like 'participants reported that...' or 'respondents felt that...' need to be inserted, to avoid statements like 'researchers are obligated to provide such feedback' - which are not established by the evidence here. Similarly, where you say 'including the fact that it helps individuals to know what to do in their life' - this isn't a fact, it is a perception. 'It can guide the medical professionals and scientists to search for treatment' - it MAY do this, but it depends on a lot of factors. Language is too strident in places.

* need to define 'ancestral information' - unclear if you are referring to maternity/paternity, inheritance patterns, population genomics or something else. There was a lot of focus on (non) paternity, which is a risk in some research projects but certainly not all, and this distinction was not made clear. Similarly, there seemed to be an assumption that everyone would have some health information come out of genomic research participation, which is a long way from the truth.

* the section where participants were described as concerned about the cost of genomic research was unclear - research participants would not normally bear these costs, nor should participation be seen as desirable for access to genomic testing (potentially undermining free consent and giving a deceptive view of the information/benefit to be gained from participation). The kind of feedback the participants seemed to envisage is very much not what most studies would provide - the finding that this is an expectation could be very useful for designing ethical guidelines, but this is undermined a bit if actually the participants just weren't given accurate information to begin with.

* privacy and confidentiality discussion would be strengthened if discussed within the context of legal obligations in this area - esp in the discussion, where it's not clear whether the problems identified with confidentiality are easily accommodated within existing laws/regulatory frameworks or not

* the variability in attitudes to feedback (to individuals/families/wider community) indicate that working out preferences for this should be part of the consent process

* the discussion is also light on - there is a lot in the results that could usefully be contextualised, but the discussion needs significant revision (unclear what you mean by 'this type of feedback of results', 'the need to know one's health condition' etc) - the strategies suggested were a bit meaningless (complying with ethical and legal requirements like privacy is not really a 'strategy') and didn't make the most of the really interesting data you've collected

* 'revealed recall bias about their participation in the concerned research study' - what do you mean by this?

* there are important implications for how we conceive of researcher obligations, benefit sharing, and capacity building priorities, but these are inadequately drawn out here

* would be good to see community research agreements discussed as part of the community engagement discussion

* acknowledgement of the familial implications (and even for ethnic groups/communities) of genomics, in addition to caring responsibilities as a reason to discuss findings with the community (not individual health data though! more general research findings)

* if the point of this project is to inform guideline development, this should be mentioned in the introduction, not the closing paragraph.

* the limitations mentioned are extremely significant. Adding some comment on how these were managed/mitigated or how they impacted (or didn't) on the results would be useful.

* the discussion needs to analyse and interpret the findings with more nuance/attention to context/thoughtfulness to be useful, given the extent of the literature in this area. Closing with 'Privacy and confidentiality, benefits, risks as well as implications for sharing need to be considered...' adds nothing to the literature and wastes the data you've collected.

Minor:

* Too many novel acronyms, none of which will be used in another paper and which only serve to distract the reader. Just write in full.

* Ownership of DNA - do you mean the data? or the actual sample? bit odd to refer to the DNA rather than the data or sample.

* some awkward phrasing throughout - revise for clarity

* 'They were assisted by eight research assistants including four females proficient in local languages.' Females should be women, and this sentence is ambiguous - are you saying the four men were not proficient in local languages?

* not sure 18-35 can be described as youth - 'younger participants' maybe?

* 'if one has a 50% chance of developing' - why 50%? do you just mean if risk is quantifiable?

* contractions (eg it's) should be eliminated

* paragraphs in the discussion are way too long

* legislated laws is a tautology

* 'Since individuals in the communities are acceptable...' this sentence makes no sense - also it is not clear why it is up to the regulators to sort it out - are current regulations inadequate? this has not been mentioned, let alone established.

6. PLOS authors have the option to publish the peer review history of their article (what does this mean?). If published, this will include your full peer review and any attached files.

Reviewer #1: No

Reviewer #2: No

Reviewer #3: No

---

## [Author Response · Author response to Decision Letter 0]

30 Aug 2022

A point by point response to review comments has been attached.

---

## [Decision Letter · Decision Letter 1]

12 Sep 2022

PONE-D-22-10220R1Feedback of Individual Genetic and Genomics Research Results: A Qualitative Study Involving Grassroots Communities in UgandaPLOS ONE

Dear Dr. Ochieng,

Thank you for submitting your manuscript to PLOS ONE. After careful consideration, we feel that it has merit but does not fully meet PLOS ONE’s publication criteria as it currently stands. Therefore, we invite you to submit a revised version of the manuscript that addresses the points raised during the review process.

The article has been greatly implemented; however, another revision step is required. Minor formal revisions are indicated in the comments below.

We look forward to receiving your revised manuscript.

Kind regards,

Andrea Cioffi

Academic Editor

PLOS ONE

Journal Requirements:

Reviewers' comments:

Reviewer's Responses to Questions

**Comments to the Author**

1. If the authors have adequately addressed your comments raised in a previous round of review and you feel that this manuscript is now acceptable for publication, you may indicate that here to bypass the “Comments to the Author” section, enter your conflict of interest statement in the “Confidential to Editor” section, and submit your "Accept" recommendation.

Reviewer #1: All comments have been addressed

Reviewer #3: (No Response)

2. Is the manuscript technically sound, and do the data support the conclusions?

Reviewer #1: Yes

Reviewer #3: Yes

3. Has the statistical analysis been performed appropriately and rigorously? 

Reviewer #1: N/A

Reviewer #3: N/A

4. Have the authors made all data underlying the findings in their manuscript fully available?

Reviewer #1: Yes

Reviewer #3: Yes

5. Is the manuscript presented in an intelligible fashion and written in standard English?

Reviewer #1: Yes

Reviewer #3: Yes

6. Review Comments to the Author

Reviewer #1: (No Response)

Reviewer #3: This is a much improved version, but some issues remain. In the clean version, there are still instances where the data analysis is described as 'content analysis' (eg in the abstract), but elsewhere it is claimed to be thematic analysis. The method used here would be more convincing if the authors cited some literature relevant to the method they actually used.

Expression is clearer throughout, but there are still errors remaining - a thorough proofread/edit would still be valuable.

Where the issue of sharing results with family members and the wider community is introduced, it would be helpful to explain why this might be done (to be honest, I cannot think of a single example of individual genomic results being given to the wider community by the research team, nor why this would ever be done - on the other hand, communicating general study findings [not individual results] to the participating community is very common and often expected - but this distinction is not raised here). In the discussion, you allude to sharing of results with family and the community as being relevant when others may also be at risk of carrying the variant of concern - this needs to be made clearer, as well as noting that this type of 'sharing' does not necessarily entail revealing individual personal information (it's more in the nature of general study findings). Clearly defining different types of 'results' would have helped with this confusion.

The paragraphs in the discussion are still really long. You raise the Data Protection and Privacy Act of 2019 and describe it as adopting the western approach without explaining what that is - this section needs a bit more clarity

7. PLOS authors have the option to publish the peer review history of their article (what does this mean?). If published, this will include your full peer review and any attached files.

Reviewer #1: No

Reviewer #3: No

---

## [Author Response · Author response to Decision Letter 1]

6 Oct 2022

A point by point response to review comments has been uploaded

---

## [Editor Report · Decision Letter 2]

11 Oct 2022

Feedback of Individual Genetic and Genomics Research Results: A Qualitative Study Involving Grassroots Communities in Uganda

PONE-D-22-10220R2

Dear Dr. Ochieng,

We’re pleased to inform you that your manuscript has been judged scientifically suitable for publication and will be formally accepted for publication once it meets all outstanding technical requirements.

Kind regards,

Andrea Cioffi

Academic Editor

PLOS ONE